# A modelling assessment of short- and medium-term risks of programme interruptions for *gambiense* human African trypanosomiasis in the DRC

Ching-I Huang[ORCID][1,2☯]*, Ronald E. Crump[ORCID][1,2], Emily H. Crowley[ORCID][1,2], Andrew Hope[3], Paul R. Bessell[4], Chansy Shampa[5], Erick Mwamba Miaka[5], Kat S. Rock[ORCID][1,2☯]

**1** Zeeman Institute for System Biology and Infectious Disease Epidemiology Research, The University of Warwick, Coventry, United Kingdom, **2** Mathematics Institute, The University of Warwick, Coventry, United Kingdom, **3** Liverpool School of Tropical Medicine (LSTM), Liverpool, United Kingdom, **4** Independent Consultant, Edinburgh, United Kingdom, **5** Programme National de Lutte contre la Trypanosomiase Humaine Africaine (PNLTHA), Kinshasa, Democratic Republic of the Congo

☯ These authors contributed equally to this work.
* ching-i.huang@warwick.ac.uk

**Data Availability Statement:** This study was a re-analysis of the *gambiense* human African trypanosomiasis (HAT) data that were obtained

## Abstract

*Gambiense* human African trypanosomiasis (gHAT) is a deadly vector-borne, neglected tropical disease found in West and Central Africa targeted for elimination of transmission (EoT) by 2030. The recent pandemic has illustrated how it can be important to quantify the impact that unplanned disruption to programme activities may have in achieving EoT. We used a previously developed model of gHAT fitted to data from the Democratic Republic of the Congo, the country with the highest global case burden, to explore how interruptions to intervention activities, due to e.g. COVID-19, Ebola or political instability, could impact progress towards EoT and gHAT burden. We simulated transmission and reporting dynamics in 38 regions within Kwilu, Mai Ndombe and Kwango provinces under six interruption scenarios lasting for nine or twenty-one months. Included in the interruption scenarios are the cessation of active screening in all scenarios and a reduction in passive detection rates and a delay or suspension of vector control deployments in some scenarios. Our results indicate that, even under the most extreme 21-month interruption scenario, EoT is not predicted to be delayed by more than one additional year compared to the length of the interruption. If existing vector control deployments continue, we predict no delay in achieving EoT even when both active and passive screening activities are interrupted. If passive screening remains as functional as in 2019, we expect a marginal negative impact on transmission, however this depends on the strength of passive screening in each health zone. We predict a pronounced increase in additional gHAT disease burden (morbidity and mortality) in many health zones if both active and passive screening were interrupted compared to the interruption of active screening alone. The ability to continue existing vector control during medical activity interruption is also predicted to avert a moderate proportion of disease burden.

from the WHO HAT Atlas and are subject to a data sharing agreement. Interested parties should apply to the WHO (contact J.R. Franco, francoj@who.int) in order to gain access to these data. Code and simulated data derived through the re-analysis undertaken in this study are available in Open Science Framework at https://osf.io/gur7c/.

**Funding:** This work was supported by the Bill & Melinda Gates Foundation (www.gatesfoundation. org) through the Human African Trypanosomiasis Modelling and Economic Predictions for Policy (HAT MEPP) project [OPP1177824 and INV-005121] (C.H, R.E.C, E.H.C, K.S.R.), through the NTD Modelling Consortium [OPP1184344] (K.S. R.), and through the TrypElim-Bandundu project [OPP1155293] (A.H., P.R.B, C.S, E.M.M.). Under the grant conditions of the Foundation, a Creative Commons Attribution 4.0 Generic License has already been assigned to the Author Accepted Manuscript version that might arise from this submission. The funders had no role in study design, data collection and analysis, decision to publish, or preparation of the manuscript.

**Competing interests:** The authors have declared that no competing interests exist.

## Author summary

Whilst the COVID-19 pandemic has produced wide-spread disruption for many disease programmes there are also a range of other factors that continue to risk programme interruptions including other disease outbreaks (e.g. Ebola, cholera, yellow fever, and measles) and the potential for political instability. In this study we examine the impact of interruptions by external factors to the *gambiense* human African trypanosomiasis (gHAT, sleeping sickness) elimination programme of the Democratic Republic of the Congo, a country which has the highest global case burden. We use our previously fitted gHAT model to simulate how transmission dynamics might be impacted by disruption to medical interventions and (where relevant) vector control activities in 38 regions within Kwilu, Mai Ndombe and Kwango provinces. For each of the six interruption scenarios we use the model to forecast case numbers and disease burden as well as estimating the expected years and probabilities of elimination of transmission. This analysis provides invaluable insight into the impact that interruptions of any persuasion could have on burden, case reporting and time to achieve elimination of transmission of gHAT in the Democratic Republic of the Congo.

## Introduction

The last two years have been particularly challenging for public health systems throughout the world due to the introduction and spread of a novel type of coronavirus (SARS-CoV-2), first identified in January 2020, which led to the coronavirus disease (COVID-19) pandemic. As well as the direct impact of COVID-19, wide-ranging and temporally varying restrictions to limit transmission of SARS-CoV-2 have meant other health programmes have been interrupted [1–4]. Whilst the pandemic has produced wide-spread disruption for many disease programmes, there are a range of other factors that continue to risk programme interruptions, particularly in low- and middle-income countries—for example in 2020 the Democratic Republic of the Congo (DRC) was also battling with outbreaks of Ebola, cholera, yellow fever and measles as well as political instability in parts of the country [5]. In this study we will examine the impact of interruptions caused by external factors on the *gambiense* human African trypanosomiasis (gHAT, sleeping sickness) elimination programme of the DRC.

gHAT is a deadly vector-borne, neglected tropical disease (NTD) found in West and Central Africa. It is caused by the parasite *Trypanosoma brucei gambiense* and transmitted through the bite of infected tsetse. The World Health Organization's (WHO) Global Health Observatory Data Repository shows a continued decline in the number of reported gHAT cases since the early 2000s [6]. In 2009 the number of gHAT cases reported globally dropped below 10,000 for the first time since the most recent epidemic started in the 1970s, and in 2019 only 876 cases were reported. In 2012, the WHO targeted gHAT for elimination as a public health problem (EPHP) by 2020 [7], with the subsequent goal being the elimination of transmission (EoT) to humans by 2030 [8].

In a recent WHO publication on monitoring the elimination of gHAT and based on data up to 2020 [9], the global EPHP targets are not yet considered fully achieved (there were fewer than 2,000 gHAT cases/year but only 83% reduction in areas at moderate or higher risk of gHAT compared to the 2000–2004 baseline). Up to the end of 2022, five countries (Togo in June 2020, Côte d'Ivoire in December 2020, Benin in November 2021, Uganda in April 2022, and Equatorial Guinea in June 2022) have received official validation for the EPHP of gHAT [10–12]. However there are still 19 out of 24 countries that reported gHAT cases that had yet

to meet the country level EPHP target (i.e. reporting < 1 case/10,000 inhabitants/year in each health district). One of these countries is the DRC which has consistently reported the highest case numbers since the early 1980s. Despite the falling number of reported gHAT cases (from 5,590 cases in 2011 to 425 in 2021), the DRC accounted for nearly 70% of all global cases before the COVID-19 pandemic started (613 out of 876 cases in 2019) [6]. gHAT is an "innovative and intensified disease management" NTD, meaning large-scale use of treatments, like mass drug administration, are not currently feasible in the management of this disease. Formerly all cases of gHAT required a lumbar puncture to determine the correct course of treatment—both treatments required intravenous administration and in-patient hospitalisation [13]. Medical advances have meant treatment has recently become simpler, and an oral drug course of fexinidazole is now approved for most patients [14, 15]. To combat the disease, three major interventions have been coordinated by the national sleeping sickness control programme in the DRC (Programme National de Lutte contre la Trypanosomiase Humaine Africaine, PNLTHA-DRC) including active screening (AS), performed by mobile teams in endemic villages to identify infected people, passive screening (PS), used to diagnose patients presenting at health facilities due to disease symptoms, and treatment of identified cases. Additionally vector control (VC) was added in selected, high-endemicity regions since 2015 through the deployment of Tiny Targets in tsetse habitat [16]. This selection of interventions, including their strengthening based on new diagnostics, drugs and technology since 2000, have led to > 95% gHAT case reduction in the DRC in the past 20 years (from 16,951 in 2000 to 425 in 2021) [6] and transmission modelling estimated an ∼84% reduction in new annual gHAT infections in the DRC (from 19,885 in 2000 to 3,130 in 2016) [17].

On 10 March 2020, the first confirmed case of COVID-19 was reported in the DRC—a Congolese citizen returning from France—one day before the declaration of the COVID-19 pandemic on 11 March. The DRC government responded quickly to its COVID-19 outbreak by announcing a national lockdown and travel bans on 28 March 2020 to stop further spread of COVID-19. As a consequence the normal operations of PNLTHA-DRC were restricted. One of the biggest changes was to the number of people tested in AS, due to the national restrictions of the government and the advice by the WHO on 1 April 2020 to temporarily halt active case finding activities [18]. Roll out of fexinidazole started in 2020, however this slowed due to the COVID-19 pandemic which restricted the capacity to provide necessary training to medical staff across the country. Likewise, expansion of VC activities to new regions was delayed. At the beginning of the pandemic it was not clear how COVID-19 would impact people's willingness or ability to attend PS, with the potential for substantially reducing detection capacity, akin to that observed during the West African Ebola outbreak in 2013–16 which saw the gHAT programmes in Guinea experience disruption of passive and active medical activities [19].

In this study, we use a dynamic transmission modelling approach to evaluate the impact of intervention interruptions on control and elimination of gHAT in the DRC at a "health zone" level (administrative regions of approximately 150,000 people). A similar question was posed and explored in a previous modelling comparison study [20], however the present analysis differs in two key ways: (1) rather than focusing on illustrative "moderate risk" health zones, we focus on 38 specific health zones in Kwilu, Mai Ndombe and Kwango provinces, accounting for variable historical screening activities and case incidence, and (2) we also explore the impact of interruptions to VC activities as well as medical interventions. This analysis does not attempt to model the scale or spread of SARS-CoV-2 itself nor other potential causes of such an interruption, rather it considers interruptions to gHAT interventions which could be plausible due to outbreaks from a variety of diseases, political unrest, natural disasters, or temporary loss of funding.

### Research objectives

Although we believe PNLTHA-DRC is now operating similarly to before the pandemic, this study aims to quantify the possible impact that different length and severity of interruptions could have on control and elimination of gHAT in the DRC. We hypothesise that there would have been some impact on transmission and case reporting during the interruption period, but it is unclear how big an impact interruption would have given the slow progressing nature of gHAT. In this paper, we approach this question independently for each of the health zones in Kwliu, Mai Ndombe and Kwango provinces via the following steps:

- Defining plausible interruption scenarios as combinations of interruptions to gHAT interventions in AS, PS, and/or VC corresponding to three categories of restrictions (which we denote as *No AS*, *No AS and reduced PS*, and *No AS or VC and reduced PS*) over a period of 9 or 21 months.

- Applying our previously fitted gHAT model [17] to simulate the transmission dynamics under the baseline (continuation of current interventions) and six interruption scenarios (interruption starts from April 2020 and interventions resume after interruption).

- Forecasting annual case reporting and yearly/accumulated disability-adjusted life years (DALYs) for the baseline and six interruption scenarios.

- Utilising yearly estimates in underlying new infections to predict expected year and yearly probability of EoT and map out the potential delay in achieving the WHO EoT goal.

## Methods

### Study areas

The operational planning of vertical interventions against gHAT including AS and VC (see below) in the DRC takes place at the coordination level. There are 11 coordinations in the DRC covering all historical endemic areas. A coordination is composed of a collection of neighbouring health zones depending on their sizes and operational accessibility. We picked Bandundu Nord and Sud coordinations (which are the same geographical region as Kwilu, Mai Ndombe and Kwango provinces) as our study areas because these regions see some of the highest levels of on-going case reporting across the country. Furthermore, there are also health zones in these coordinations with low case reporting, and therefore we expected to see a range of impacts of interruptions across these different prevalence settings. A map showing the location of Bandundu Nord and Sud within the DRC and their relation to Kwilu, Mai Ndombe and Kwango provinces can be found in S1 Text: Fig G. Despite the smallest intervention area being at village level, our simulations are carried out at health-zone level due to data availability and associated computational challenges. The methods used in this paper could readily be applied to health zones in other coordinations of the DRC using our existing model fits or foci in other gHAT endemic countries if existing fits are available e.g. the Mandoul focus of Chad [21] or health districts in Côte d'Ivoire [22], or following new fitting to historical data.

### Current interventions

In this modelling study we simulated the impact of interruptions to on-going strategies designed to reduce gHAT transmission and burden. Strategies against gHAT are comprised of several key components which act in different ways to curtail disease and transmission of the parasite.

AS is a form of mass screening of individuals living in "at risk" villages, currently using either the card agglutination test for trypanosomiasis (CATT) or by rapid diagnostic tests (RDTs); the choice of screening test can depend on the size of the mobile team—traditional truck-based teams have capacity to use a generator which makes the CATT test feasible, whereas mini-mobile teams which travel to hard-to-reach villages by motorbike utilise RDTs [23]. Following a positive CATT or RDT, visual confirmation of the parasite in blood or lymph node aspirate must be made via microscopy to confirm infection. In the health zones from Bandundu Nord and Sud coordinations included in this analysis, annual coverage of AS ranged from 0–85% (median 13%) during the period 2014–2018. As per the WHO guidance [13], villages that have not reported cases for 3 years are generally not screened so health zones which more recently reporting cases typically have higher AS coverage. Since 2018 mobile teams in Bandundu have been equipped with tablets to enable video recording of microscopy for quality assurance. Our fitted model explicitly uses the number of people tested during AS for 2000–2016, which captures the historical variation in AS coverage. Despite the potential for decreasing interest in attending AS when knowledge of disease is diminishing, data shows that the maximum number of people screened occurred after 2016 in 19 of out 38 health zones in Bandundu Nord and Sud coordinations. Our future projections use the average of known 2014–2018 AS coverage (the 2014–2016 HAT Atlas and the 2017–2018 PNLTHA-DRC AS data) for each health zone as the future coverage to reflect the observed trends in participation in recent years.

PS is a form of screening integrated in fixed health facilities in gHAT endemic regions. It relies on infected people with symptoms seeking out medical care in facilities that have gHAT RDTs available and trained staff to recognise symptoms of this increasingly uncommon disease. Access to such facilities can be geographically heterogeneous, and in the case where no microscopy is available in the screening facility, if an initial RDT is positive, further travel is needed to enable the suspect to have infection confirmed. This can result in high levels of attrition between screening and confirmation [24]. As self-presentation by infected people is driven by symptoms, people identified through PS are typically in late-stage (stage 2) disease, meaning that they are not only suffering from higher disease burden, but there is a longer duration in which tsetse can acquire infection from them and create additional onward infections. Some infected people may never receive a diagnosis, especially if PS is weak near their residence, and would therefore die (unreported) from disease-induced mortality. Despite the challenges with PS, 41% of gHAT cases in Bandundu Nord and Sud coordinations were diagnosed through the passive system in 2012–2016, and improvements to the PS system in Bandundu between 2000–2016 are believed to have contributed to the decline in gHAT transmission and proportionally fewer deaths following stage 2 infection [17].

Treatment for gHAT occurs following confirmation of a case either by AS or PS. Prior to 2020, "staging" of the disease, via lumbar puncture, was also required to determine whether or not the parasite had crossed the blood-brain barrier. For both stages hospitalisation was required for treatment, with stage 1 disease being treated with pentamidine, and stage 2 disease being treated with nifurtimox-eflornithine combination therapy (NECT) [13]. Since 2020 a new oral drug, fexinidazole, has been rolled out in the DRC which enables treatment of either stage in most cases (severe gHAT, children under six or those with a weight of less than 20kg, unfortunately still require staging and intravenous drugs) [14].

VC is an intervention which aims to reduce tsetse numbers sufficiently in order to interrupt transmission of a disease to humans. For gHAT in the DRC, one of the major VC methods conducted is the deployment of Tiny Targets, an insecticide-treated fabric mounted on sticks placed near riverine-forest habitats. Large-scale implementation of VC in the DRC began in mid-2015 in Yasa Bonga health zone. This was further expanded to five health zones by 2019,

with plans for an additional six health zones in 2020 and 2021. Deployments are carried out by the provincial-level VC teams and supported by both PNLTHA-DRC and Liverpool School of Tropical Medicine (LSTM) [25]. Previous studies have shown that the effectiveness of VC (measured by the annual reduction in tsetse population density) is between 80 and 99% [16, 26–28]. Based on reported VC effectiveness, modelling exercises predict a rapid decline in underlying new infections when VC is used to supplement medical interventions and that EoT is expected to be achieved within 4 years of starting VC roll out [29].

## Model

In the present study we used our mechanistic, deterministic gHAT model previously fitted at the health-zone level [17] to perform projections under the baseline strategy and six interruption scenarios. Here we give a brief synopsis of the model, however, a full description of the model equations is given in S1 Text and in the original model fitting article [17]. This model has been developed and refined over many years in conjunction with PNLTHA-DRC and a variety of other stakeholders to ensure the modelling team fully understand historical data and context, capture the known epidemiology, and account for region-specific interventions within the model framework. We give a more specific list of stakeholder inputs and interactions in S3 Text.

Our model is a deterministic, compartmental model explicitly including humans and tsetse to simulate the infection dynamics. We account for different infection states and disease progression in humans with a SEIIR (susceptible-exposed-infected (stage 1)-infected (stage 2)-recovered) formulation which aligns with our knowledge about the natural history of gHAT disease. For the tsetse population we include standard susceptible ($S_V$) and infectious ($I_V$) compartments, but additionally we include (i) a compartment for the number of pupae ($P_V$), (ii) three compartments for the extrinsic incubation period (EIP) in tsetse which employs the "linear-chain trick" to force a gamma-distributed EIP, and (iii) a "susceptible but non-teneral (fed)" compartment ($G_V$) which enables us to capture a specific tsetse-trypanosome biological phenomenon that following a tsetse's first blood meal it has reduced susceptibility to infection by *T. b. gambiense* trypanosomes.

Transmission in the model occurs via standard criss-cross interactions expected for vector-borne infections—infection from humans to flies, or flies to humans. We do assume that flies may take blood meals on non-human animals, but that (in this study) does not result in non-human animal infections. Previous development of this model has found that heterogeneity in the risk of people being bitten is important to be able to replicate time-series trends observed in reported case data for different settings [21, 26, 30], and furthermore the best fitting models had high-risk people that did not participate in AS, which also aligned with anecdotal evidence. In the present study we therefore consider that our low-risk group of people participate in AS randomly and according to the coverage of annual AS, but our high-risk group do not participate and can only be detected and treated by seeking help through the PS system in fixed health facilities. We assume that high-risk and low-risk people have the same detection rate in PS if they are not found through AS, and that most infections found through PS have progressed to stage 2 disease.

There are two types of parameters in our model, fixed and fitted parameters. Most fixed parameters are biological parameters, and therefore their values are most likely to be location-independent (health zones in our study) with less uncertainty. Fitted parameters, on the other hand, are believed to vary from location to location with much bigger uncertainty so we fit these on a health zone by health zone basis. A full sensitivity analysis performed elsewhere for this same model [31] used a ranked partial correlation method and found that the fitted

parameters were the ones leading to most variation in outcomes, strengthening our justification for which parameters should be fixed and fitted.

The fitting performed previously [17] used 2000–2016 WHO HAT Atlas case data [32] accounting for varying AS coverage by location and year. Using an adaptive Metropolis-Hastings Markov chain Monte Carlo method we estimated geographically varying parameters to provide health-zone-specific fits. The fitted model also takes into account previous advances in AS, PS, and VC in 2000–2016 period: this includes (1) a video recording confirmation system, which we assume results in perfect specificity in the AS algorithm, has been used in active case diagnostics in Mosango and Yasa Bonga from 2015, (b) reduced time to detection through PS, and (c) biannual VC in Yasa Bonga (began in mid-2015) with an effectiveness of 90% tsetse reduction after one year. S1 Text provides complete parameter descriptions (Tables A and B), model equations for improved PS (Equation (2)), and tsetse mortality due to VC (Equation (4) and Fig D). Although fexinidazole roll out began in 2020, if it is implemented according to the current WHO guidelines—which still requires parasitological confirmation—we assume this has no meaningful impact on the way we model transmission or case reporting in the present study; i.e. we assume that the rate of detection followed by treatment either via AS or PS is not altered by switching to fexinidazole.

The posterior parameter sets for each health zone are then used to project forwards in time, applying various intervention settings described below (baseline strategy and interruption scenarios). In this study, all interruption scenarios occur during this projection phase and no actual data other than the timing of the start of the COVID-19 epidemic in the DRC was used in our projections.

Outputs from the model include expected annual active and passive case reporting, numbers of new infection events each year, and annual DALYs as a measure of disease burden. For gHAT, most of the DALYs arise from disease-induced mortality from untreated individuals and a smaller proportion is morbidity from the number of person-years lived with either stage 1 or stage 2 disease [33]. The death rate from gHAT—and subsequently the expected number of deaths per year—is estimated by the model as part of the fitting process, where we use (where available) trends in the number of staged active and passive cases and duration of stage 1 and 2 to infer the (undetected) death rate following stage 2 disease; more details are given in Crump et al. [17] and in S1 Text. Our underlying new infections are similarly estimated during the model fitting process by matching model case outputs to the WHO case data. Since there is no way to directly measure annual new infections, this model output is used to compute the probability that EoT has been met.

**Baseline strategy (no interruption).**   In previous projections, we simulated the impact of four future strategies from 2017 onward without interruptions for 168 health zones of the DRC [29]. The strategies comprised of different levels of AS with or without additional VC. In the present study, we have updated our projections to include additional VC roll out which occurred between 2015–2019 or was planned for 2020 or after (see Table 1 for a list of health zones and roll out years). For those health zones without VC history or future plans, the baseline strategy used was *MeanAS* representing a mean coverage of AS with the mean number of people screened in each health zone being calculated based on the 2014–2016 HAT Atlas and the 2017–2018 PNLTHA-DRC AS data. For those health zones with either VC history or future VC plans, the baseline strategy is *MeanAS+VC*. As data on tsetse reductions has not yet been reported for health zones where it has recently begun, a fixed effectiveness of 80% tsetse reduction in one year is assumed in the *MeanAS+VC strategy*; this is in line with lower-end estimates reported elsewhere [16, 26–28]. In Yasa Bonga we use the reported effectiveness of 90% [16]. Video confirmation of parasitological diagnosis is assumed in the AS algorithm from 2018 to remove false positives in active cases, in line with tablet roll out by

**Table 1. Baseline strategy by health zone.**

| Health zone | Baseline strategy | AS coverage* | VC status† | VC effectiveness |
|---|---|---|---|---|
| Yasa Bonga | MeanAS+VC | 57% | Since 2015 | Reported as 90% [16] |
| Masi Manimba | MeanAS+VC | 46% | Since 2018 | Assumed as 80% |
| Bandundu | MeanAS+VC | 34% | Since 2019 | Assumed as 80% |
| Kikongo | MeanAS+VC | 21% | Since 2019 | Assumed as 80% |
| Kwamouth | MeanAS+VC | 48% | Since 2019 | Assumed as 80% |
| Bokoro | MeanAS+VC | 27% | Planned for 2021 | Assumed as 80% |
| Bolobo | MeanAS+VC | 48% | Planned for 2020 | Assumed as 80% |
| Bulungu | MeanAS+VC | 17% | Planned for 2020 | Assumed as 80% |
| Mokala | MeanAS+VC | 20% | Planned for 2021 | Assumed as 80% |
| Mushie | MeanAS+VC | 29% | Planned for 2021 | Assumed as 80% |
| Yumbi | MeanAS+VC | 15% | Planned for 2021 | Assumed as 80% |
| Mosango | MeanAS | 34% | No | 0% |
| Elsewhere | MeanAS | Varies | No | 0% |

AS: active screening; VC: vector control

*Based on 2016 estimated populations

†All initial VC deployments were started, or were planned to be started mid-way through the calendar year, so that in the first year there is a single deployment but two deployments per year afterwards.

PNLTHA-DRC in Bandundu Nord and Sud coordinations. We also assumed the detection rates in PS remain the same as 2018 from 2019 onward. The baseline strategy name, AS coverage, VC status and VC effectiveness under the baseline strategy for each health zone are summarised in Table 1.

**Interruption scenarios.** In this study we examine interruptions to gHAT interventions which could be caused by an outbreak of other diseases, political instability or other factors. Different modes of disruption vary substantially, however we did not want to concretely link specific causes of interruption to specific changes in strategy implementation. Therefore our analyses are framed in the context of interruption to different intervention components, to determine what type and what severity of impact interruptions could have, regardless of cause. Motivated by the COVID-19 pandemic and interruption durations that were considered plausible in early 2020, we focus on interruptions starting in April 2020 lasting until either the end of 2020 or the end of 2021. Although the COVID-19 pandemic has had seemingly minimal impact on strategy—with only a reduction in the AS coverage for 2020—it was not clear at the outset of this novel pathogen outbreak whether this would have more far-reaching consequences of the type experienced during the West African Ebola outbreak in 2013–2016 where the PS system was heavily impacted in addition to the AS coverage falling [19]. Finally, political unrest could impact all intervention types, but this would be highly dependent on the nature, location, and duration of the situation. Thus, we established three categories of restrictions with different levels of impact on current interventions: *No AS* with assumed mobile screening suspension, *No AS and reduced PS* with limited health facility capacity or willingness of public to attend health facilities in addition to mobile screening suspension, and *No AS or VC and reduced PS* with mobile screening suspension, limited health facility capacity, and no VC deployment.

For scenarios including the interruption of AS, we assume some AS had already occurred in 2020 before restrictions were imposed in April. Therefore AS coverage in 2020 is a quarter of the baseline strategy. For PS we assume the impact of interruptions is partial therefore PS

**Fig 1. Gantt chart showing simulated baseline (no interruption) and interruption scenarios.** Each section shows the model assumptions for the timing of interruptions to different strategy components (interventions). The green background indicates the intervention is running as normal. Orange background indicates the intervention has been suspended (in the case of AS or VC) or reduced (in the case of PS). E.g. a 9-month interruption to AS is assumed to last from April 2020 to December 2020 and so has an orange background for this time period. The right hand side of the diagram shows how the different types of interruptions are combined together to make up the baseline and six interruption scenarios. Each health zone falls into one of four categories, (i) existing VC prior to 2020, (ii/iii) VC was planned to start in 2020/21, or (iv) no VC planned. For health zones with existing or planned VC, interruption is indicated by crosses for no deployment when there was one expected in the baseline. In these health zones there may or may not have been deployments at the beginning of 2019—we simulate the deployments that occurred in specific health zones. In regions with existing VC deployments, only the deployments during the interruption period are impacted. However, for health zones planning to start new VC intervention, we assume the initial deployment was pushed back by one or two full years and so deployment does not take place immediately after interruption ends; these additional missed deployments are denoted by crosses with stars. AS: active screening; PS: passive screening; VC: vector control; B: Baseline.

detection rates are reduced to the post-1998 level. Finally for VC interruptions, we assume that any deployments planned for July 2020 would be suspended in the 9-month scenario. Furthermore VC deployments planned for July 2020, January 2021 and July 2021 would be suspended in the 21-month scenario. We assume that, if VC deployments are interrupted, that there is some possibility for bounceback of the tsetse populations. We model the tsetse population explicitly and this bounceback is assumed to be driven by repopulation from remaining tsetse in the region (see S1 Text). In regions with planned roll out in 2020 or 2021 which are interrupted we assume these are then postponed to July 2021 and July 2022 in the 9- and 12-month scenarios respectively.

Graphical illustrations in Fig 1 outline timelines and impacts on AS, PS, and VC under six interruption scenarios together with the baseline strategy considered in this analysis.

## Results

### Health zones with no previous or planned VC

In health zones which have not had VC (inclusive of the majority of the health zones in Bandundu Nord and Sud coordinations considered in this analysis), the predicted impact of interruptions to interventions on case reporting, underlying transmission, disease burden, and probability of meeting the EoT target is dependent on several factors. These include endemicity at the beginning of 2020, coverage of AS, and the strength of PS in 2019 (especially for detection of stage 2 disease). To illustrate our findings for health zones with no previous or planned VC, we have selected Mosango health zone as an exemplar. Of note, Mosango has an estimated population size of around 125,000 in 2016, had an average of 0.99 reported cases per 10,000 people per annum during 2014–2018 (classified as a low-risk health zone according to the gHAT risk categories defined by the WHO [34]), had moderate AS coverage during 2014–

2018 (34%), and it has been estimated there was substantial improvement in detection rate for stage 2 disease between 2000–2016 (shortening the time for stage 2 detection from 393 to 293 days).

In the scenario of *No AS* (and continuing PS), our model predicts slightly larger prediction intervals on active cases after AS resumes in Mosango (first column, Fig 2). Although the increase in numbers of passive cases and new infections are marginal during and after the interruption period, the model predicts 35.2 and 76.3 additional DALYs accrued over the next 10 years in the 9- and 21-month interruption scenarios respectively. It is mainly due to the disease progression in infected people but also due to a stagnation in the reduction of underlying transmission caused by the interruption of AS. A delay in the YEoT is only predicted in the 21-month interruption scenario (delay of one year). The model predicts 2.7 and 5.9% reduction in PEoT by 2030 when AS is interrupted by 9 and 21 months respectively.

In the *No AS and reduced PS* scenario (second column, Fig 2), the model predicts more post-interruption active and passive cases and new infections compared to those in the *No AS* scenario. Fewer infections being picked up during the interruption period via PS, especially in stage 2 disease, has a substantial impact on disease burden. Reduced PS (delay in passive detection) in addition to no AS could result in doubling the additional DALYs accrued (61.4 and 152.7 additional DALYs for 9- and 21-month interruptions) compared to interruption of AS alone in Mosango. This highlights the particular importance of maintaining functional PS in managing disease burden in the absence of vertical interventions, even if it is imperfect. The estimated delay in YEoT is one year, and the PEoT by 2030 is predicted to be lower by 4.6 and 11.2% for 9- and 21-month interruptions respectively (in line with the doubling of additional DALYS accrued). Since there is no on-going or planned VC in Mosango, the scenario of *No AS or VC and reduced PS* is absent in Fig 2.

## Heath zones with existing or planned VC

We selected Kwamouth, where VC began in mid-2019, as an exemplar to study the potential impact caused by different interruptions in areas that had existing, but recently started, VC before 2020 (see Fig 3). Data show that Kwamouth health zone (8.7 reported cases per 10,000 people per annum in 2014–2018) is in the gHAT moderate-risk category [34, 35] and its average recent AS covered 48% of the population (estimated population size was around 131,000 in 2016). Our fitted model estimates that there was substantial improvement in detection rate for stage 2 disease between 2000–2016 (shortening the time for stage 2 detection from 559 to 380 days).

The projections in Kwamouth under the baseline show that case reporting would be expected to continue for some time as humans infected before the roll out of VC can take several years to be identified even though VC is expected to interrupt transmission by rapid reduction of tsetse populations. In the *No AS* scenario (first column, Fig 3), our model predicts fewer active cases during the interruption period and more active cases after AS resumes. The predicted passive cases and DALYs are slighter greater than those estimated under the baseline in 2020–2030 (also see Fig A in S2 Text). Despite having more overall active and passive cases and disease burden, the difference of predicted underlying new infections between the baseline and the interruption scenario are negligible. Therefore, PEoT remains unaffected and the model predicts virtual certainty in EoT by 2030 due to tsetse reductions from the continuation of VC in this scenario.

In the *No AS and Reduced PS* scenario (second column, Fig 3), there are fewer passive cases and a substantial drop in PEoT in 2021 (by 14.9% and 20.4% for 9- and 21-month interruptions) during the interruption period in contrast to the *No AS* scenario. However, there is no

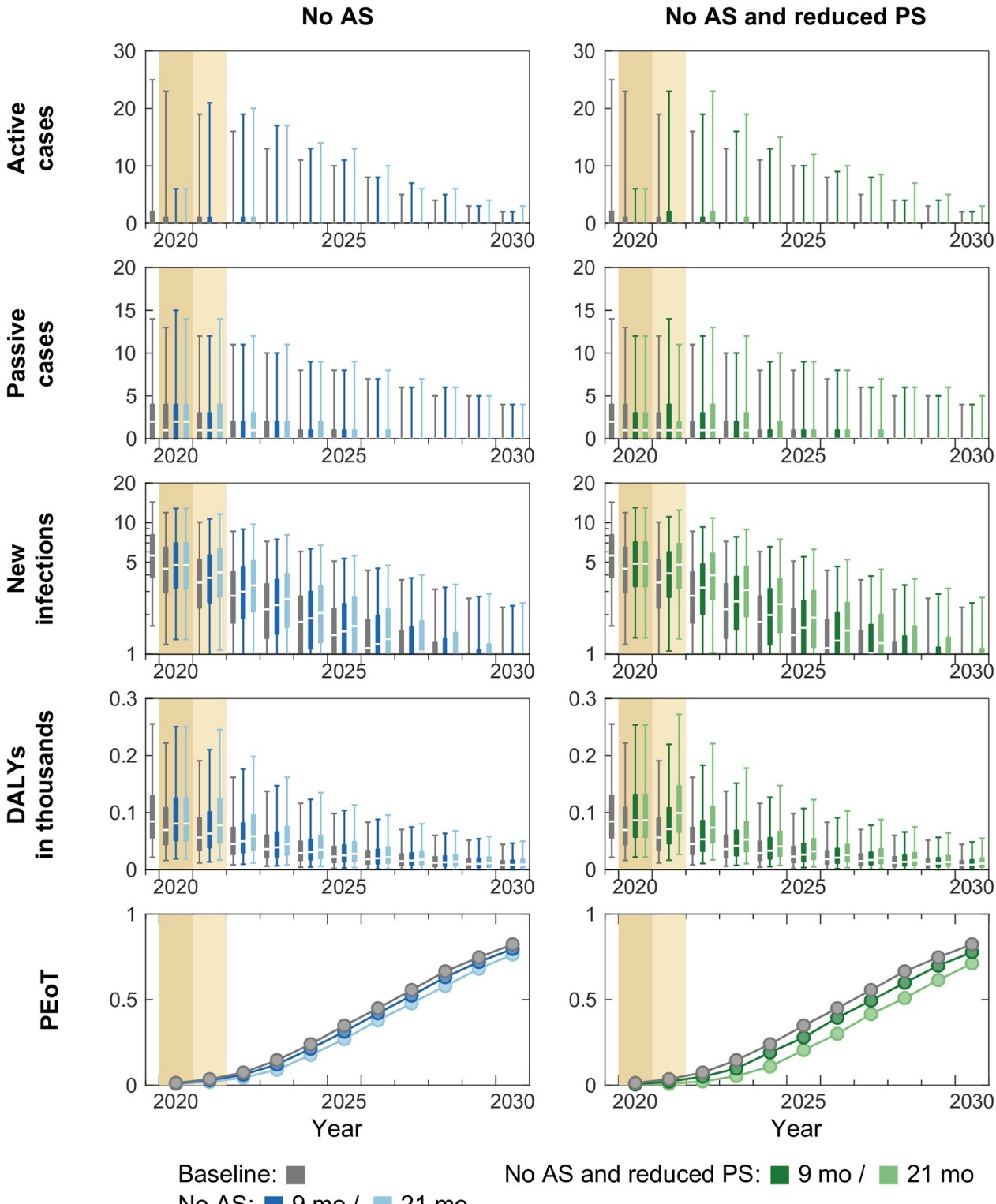

**Fig 2. Time series of model outputs in Mosango health zone (no on-going or planned VC) under the baseline and six interruption scenarios.** Projected baseline is shown in gray. Interruptions, indicated by coloured background, are assumed to take place in April 2020 and last until the end of 2020 (darker tan) or 2021 (lighter tan) in our simulations. The projections of the baseline (gray) and interruption scenarios (coloured) start from 2020. Note that the results of the "No AS or VC and reduced PS" scenario are identical to "No AS and reduced PS" because there was no on-going or planned vector control (VC) in Mosango and so is not shown. There are n = 10,000 independent samples, 10 from each of 1,000 independent

samples from the joint posterior distributions of the fitted model parameters. Box plots summarise parameter and observational uncertainty. The lines in the boxes represent the medians of predicted results. The lower and upper bounds of the boxes indicate 25th and 75th percentiles. The minimum and maximum values are 2.5th and 97.5th percentiles and therefore whiskers cover 95% prediction intervals. AS: active screening; PS: passive screening; VC: vector control; DALYs: disability-adjusted life years; PEoT: probability of elimination of transmission.

delay to the predicted YEoT. Our model results show that the reduction in passive detection rates has substantial impact on disease burden (365.9 and 727.5 additional DALYs for 9- and 21-month interruptions) in contrast to a rapid reduction in new infections when VC continues.

In the *No AS or VC and reduced PS* scenario (third column, Fig 3), new infections do not fall as rapidly as the baseline or other interruption scenarios where VC continues, and therefore the model predicts a shift on the PEoT curve for both 9- and 21-month interruptions. The value of the PEoT increases rapidly after VC resumes and all realisations achieve EoT by 2030 despite model predictions that YEoT is delayed by 1 and 2 years compared to baseline in 9- and 21-month interruptions. In Kwamouth, there was only two rounds of VC deployments prior to the start of interruption scenarios, and therefore numbers of tsetse and infected humans are comparatively high. Hence, our model predicts large amounts additional DALYs accrued (463.7 and 1069.3 additional DALYs for 9- and 21-months interruptions) in the scenarios with VC interruption.

Bokoro health zone is one of six health zones that had planned VC in 2020 or 2021 prior to 2020. The historical data show that Bokoro health zone is also at moderate risk from gHAT (3.3 reported cases per 10,000 people per annum in 2014–2018) [34, 35]. It had a population size of around 225,000 people in 2016 and recent AS covered on average 27% of the population. Previous modelling results indicate that PS is still poor despite the substantial improvement in detection rate for stage 2 disease between 2000–2016 (shortening the time for stage 2 detection from 4348 to 2500 days). Fig 4 shows that the trends and patterns of modelling outputs under the baseline strategy and interruption scenarios in Bokoro are similar to those predicted in Kwamouth. The model predicts little impact on new infections and the same PEoT curves if only AS is interrupted (first column, Fig 4). When both AS and PS are affected in the *No AS and reduced PS* scenario (second column, Fig 4), there is limited and short-term impact on transmission (i.e. PEoT lowered by 5.5 and 13.4% in 2023) and therefore no delay in the YEoT. As seen in the shift in the PEoT curve, one year delay in the YEoT is predicted in the 21-month *No AS or VC and reduced PS scenario*, and no delay is expected if the lockdown ends in 2020 (9-month interruption). This difference is due to VC being planned in 2021 in Bokoro, and therefore the VC intervention itself is unaffected in the 9-month *No AS or VC and reduced PS* scenario. The 9-month *No AS or VC and reduced PS* and *No AS and reduced PS* scenarios are therefore equivalent. Note that predicted case reporting and DALYs tail off slowly in the baseline and all scenarios because Bokoro has lower passive stage 2 detection rate.

## Comparison of delay in YEoT and DALYs across all health zones

Comparing the model outputs of interruption scenarios in Mosango, Kwamouth, and Bokoro, the patterns of case reporting and DALYs are similar. Conversely patterns of new infections and PEoT differ in the *No AS* and *No AS and reduced PS* scenarios for health zones with and without VC (Kwamouth and Bokoro vs. Mosango; see Figs 2–4). In Kwamouth and Bokoro health zones, the continued rapid decline in the numbers of new infections and the rate at which the PEoT reaches 1 irrespective of the interruption scenario is a consequence of the continuation of VC. It is expected, therefore, that a delay in the YEoT is unlikely to happen in those scenarios where VC remains uninterrupted for health zones with existing or planned VC.

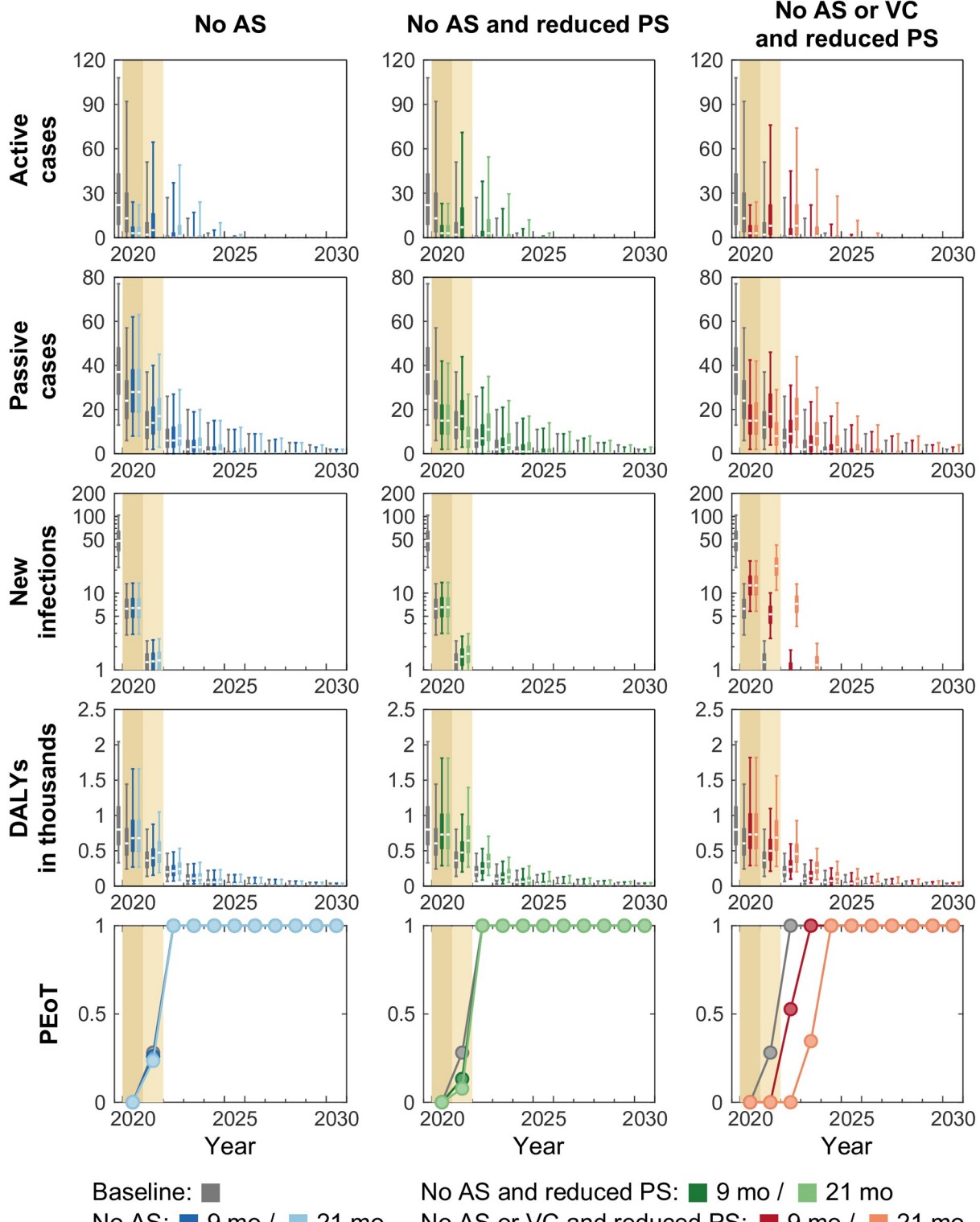

**Fig 3. Time series of model outputs in Kwamouth health zone (on-going vector control since 2019) under the baseline and six interruption scenarios.** Projected baseline is shown in gray. Interruptions, indicated by coloured background, are assumed to take place in April 2020 and last until the end of 2020 (darker tan) or 2021 (lighter tan) in our simulations. The projections of the baseline (gray) and interruption scenarios (coloured) start from 2020. There are n = 10,000 independent samples, 10 from each of 1,000 independent samples from the joint posterior distributions of the fitted model parameters. Box plots summarise parameter and observational uncertainty. The lines in the

boxes present the medians of predicted results. The lower and upper bounds of the boxes indicate 25th and 75th percentiles. The minimum and maximum values are 2.5th and 97.5th percentiles and therefore whiskers cover 95% prediction intervals. AS: active screening; PS: passive screening; VC: vector control; DALYs: disability-adjusted life years; PEoT: probability of elimination of transmission.

The maps in Fig 5 and Table A in S2 Text summarise the median delay in YEoT under different interruption scenarios in 38 modelled health zones. For health zones that have no on-going or planned VC, a delay in the YEoT is more likely in *No AS and reduced PS* scenarios with the delay being up to the length of the interruption but is unlikely in the 9-month *No AS* scenario. On the other hand, for health zones that should have VC during the interruption period, our model predicts a delay only occurs in the *No AS or VC and reduced PS* scenarios and is capped to the length of the interruption as VC is an efficient tool in reducing transmission to humans.

Fig 6 shows a stacked bar chart summarising DALYs under the baseline and additional DALYs caused by different interruption scenarios in all modelled health zones in Bandundu Nord and Sud coordinations. Looking across these health zones, a key feature of interruptions is the decline in case reporting from the impacted interventions during the interruption period (see Figs 2–4 and Figs B–AJ in S2 Text). Infected humans spend longer being infected and potentially create more new infections resulting in an overall increase in predicted DALYs between 2020–2030. The magnitude of additional DALYs accured is not only dictated by the baseline infection prevalence but also determined by the between-health-zone differences in epidemiology and medical interventions. For example, our model predicts Kwamouth has fewer total DALYs accrued in 2020–2030 than Bokoro (1445.9 and 3221.4 respectively under the baseline) but more additional DALYs accrued in both *No AS* scenarios due to higher AS coverage (48% in Kwamouth and 27% in Bokoro). This finding highlights the importance of the capacity and strength of medical interventions to gHAT management.

## Discussion

This analysis was performed to simulate the potential impact of interruptions to gHAT programmes due to a range of possible causes, however the COVID-19 pandemic was at the forefront of our concerns and consequently interruptions were simulated from April 2020 lasting either for the rest of 2020 or until the end of 2021. Fortunately, in the DRC we now know that some of our worse case interruption scenarios did not occur, and it is believed that the main impact was a reduced coverage of AS in 2020 (similar to the 9-month *No AS* interruption scenario in this article). In the analysis, mean screening in 2019 was 16% annually averaged across health zones and our simulated coverage for 2020 was 4% for all interrupted scenarios. In reality this is now known to have been 26% and 16% for 2019 and 2020 respectively. In health zones with existing VC prior to 2020, deployments continued [25], although planned roll outs in Bolobo and Bulungu health zones were delayed to 2021.

We note that direct interruptions from events such as COVID-19 can also have longer knock-on effects for programme implementation. Despite activities resuming in late 2020 there have been a range of other challenges for PNLTHA-DRC continuing into 2022, partly due to reorganisation of the local active screening teams, but also due to transitions between study projects and associated funding, and stock outs of drugs in the places they are needed. Even when there are no direct rules in place to prevent vertical activities following a short interruption, operational barriers remain one of the biggest hurdles to overcome, particularly in remote and conflict-prone regions of this huge country.

Our modelling results highlight the importance of the PS system in avoiding disease burden in the Bandundu Nord and Sud coordinations in the DRC, as noted when comparing

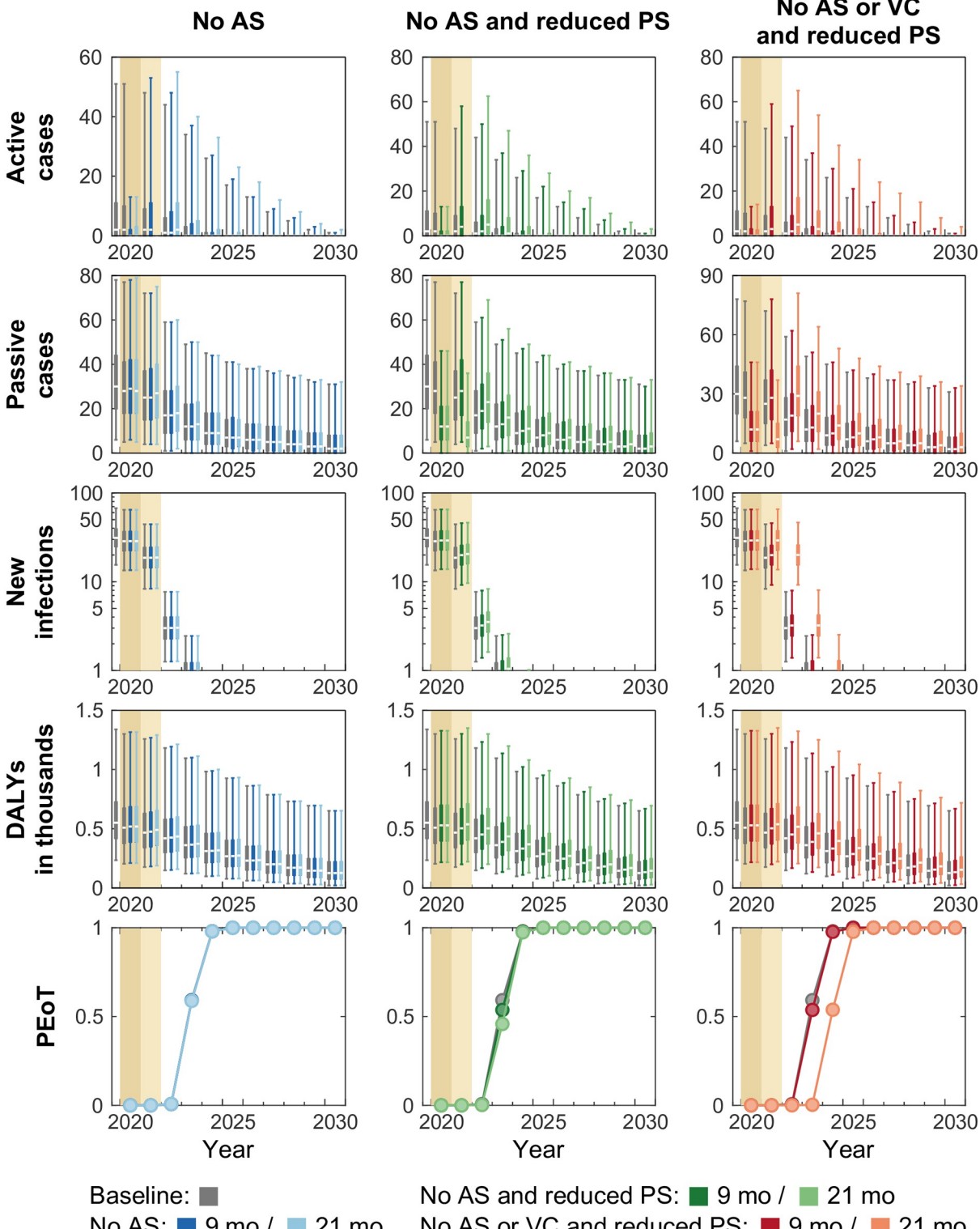

**Fig 4. Time series of model outputs in Bokoro health zone (planned vector control from 2021) under the baseline and six interruption scenarios.** Projected baseline is shown in gray. Interruptions, indicated by coloured background, are assumed to take place in April 2020 and last until the end of 2020 (darker tan) or 2021 (lighter tan) in our simulations. The projections of the baseline (gray) and interruption scenarios (coloured) start from 2020. There are n = 10,000 independent samples, 10 from each of 1,000 independent samples from the joint posterior distributions of the fitted model parameters. Box plots summarise parameter and observational uncertainty. The lines in the boxes present the

medians of predicted results. The lower and upper bounds of the boxes indicate 25th and 75th percentiles. The minimum and maximum values are 2.5th and 97.5th percentiles and therefore whiskers cover 95% prediction intervals. AS: active screening; PS: passive screening; VC: vector control; DALYs: disability-adjusted life years; PEoT: probability of elimination of transmission.

strategies with continuing PS to those with reduced PS. This in spite of challenges linked to accessibility of fixed health centres, training of health facility staff, and local availability of diagnostics and drugs [24]. As well as direct burden benefits, a robust PS system will be essential moving into the future, as when vertical interventions (AS and VC) are ceased after having no case reporting for 5 years [13], PS will remain the main mode of monitoring the success of the programme or triggering reactive vertical interventions.

More generally, this analysis demonstrates the type of results that could be expected from interruptions of 9 or 21 months due to any cause, which we believe provides valuable insights not solely into the impact of COVID-19 on gHAT in the DRC, but how other potential interruptions in the future could affect gHAT burden, case reporting and trajectory towards EoT.

## Limitations

In the following paragraphs, we discuss limitations of our study related to the model structure, assumptions, and data availability.

The model variant used here has been published and refined several times before for both the DRC and Chad [17, 21, 26, 29, 30, 36, 37]. Although stochastic models are generally better suited to capturing dynamics approaching elimination, we do not expect this variant of the deterministic model (representing average expected dynamics) to result in different conclusions from a stochastic model, as other work by this group demonstrated similar predictions generated by the analogous stochastic gHAT model [20]. In this study we chose to utilise the deterministic version as we expected limited impact on key outcomes (change in disease burden with interruptions and delay in elimination year) and to avoid the comparative high computational cost of running a stochastic model. If a similar analysis was conducted for smaller spatial scales (e.g. a health area with around a tenth of the population size of a health zone) then stochastic models would become more desirable.

This model variant does not include either non-human animal transmission nor self-curing asymptomatic human infections, which have both been flagged as a potential concern for meeting EoT [38]. Other modelling work comparing the model variant used in this study with a variant including possible animal transmission found that, in some health zones in Bandundu Nord and Sud coordinations, EoT may occur later if animal transmission is occurring [17]. Similarly, if frequent self-curing asymptomatic transmission exists, this may result in more pessimistic predictions for EoT [31]. Although the impacts of interruptions on delays to EoT and additional DALYs accrued in models with animal transmission or self-curing asymptomatic infections are unknown, we speculate that results would not be qualitatively dissimilar to those presented here. This is because (1) infection timescales are still long and (2) fitting different model variants to other regions which had a gap in AS for 2 years then resumed showed no discernible difference between inferred transmission trends for different model variants [21, 26].

Although this study did not make alterations to the transmission and reporting model used with the roll out of fexinidazole, we acknowledge the clear benefits of the new drug to patients and clinicians. Furthermore we do believe that the new drug could have more important impacts on costs and resources (including in patient hospitalisation, staff time, etc.) and should therefore be included in any cost-effectiveness analyses. Another modelling study has explored

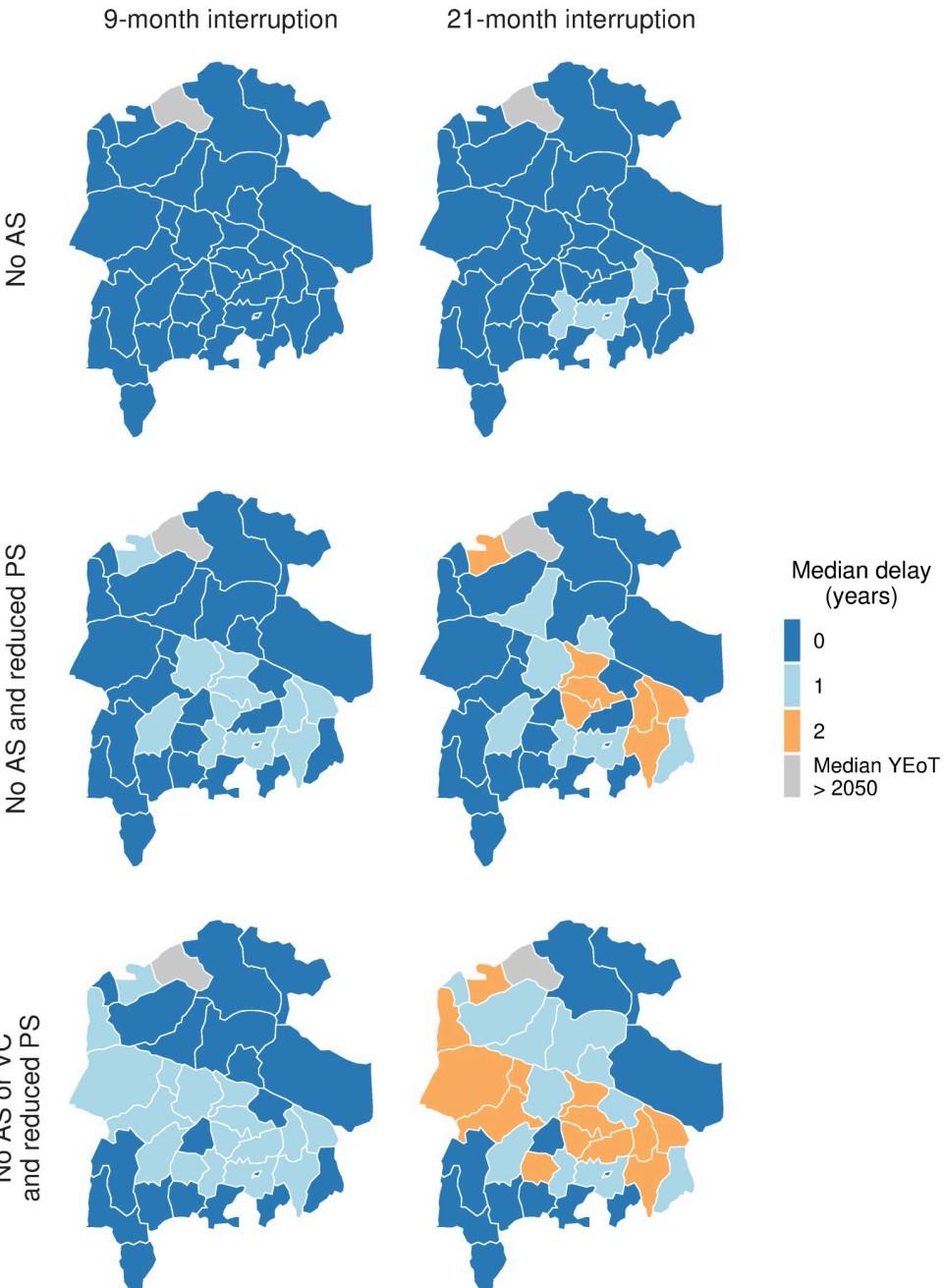

**Fig 5. Maps of median delay in YEoT under six interruption scenarios in 38 modelled health zones in Bandundu Nord and Sud coordinations.** In each health zone, the baseline strategy is either *MeanAS* or *MeanAS+VC* depending on its VC status (summarised in Table 1). 'Median YEoT > 2050' where the median year of elimination of transmission, either baseline, interruption, or both, was beyond 2050. Shapefiles used to produce these maps were provided by Nicole Hoff and Cyrus Sinai under a CC-BY licence (current versions can be found at https://data. humdata.org/dataset/drc-health-data). AS: active screening; PS: passive screening; VC: vector control; YEoT: year of elimination of transmission.

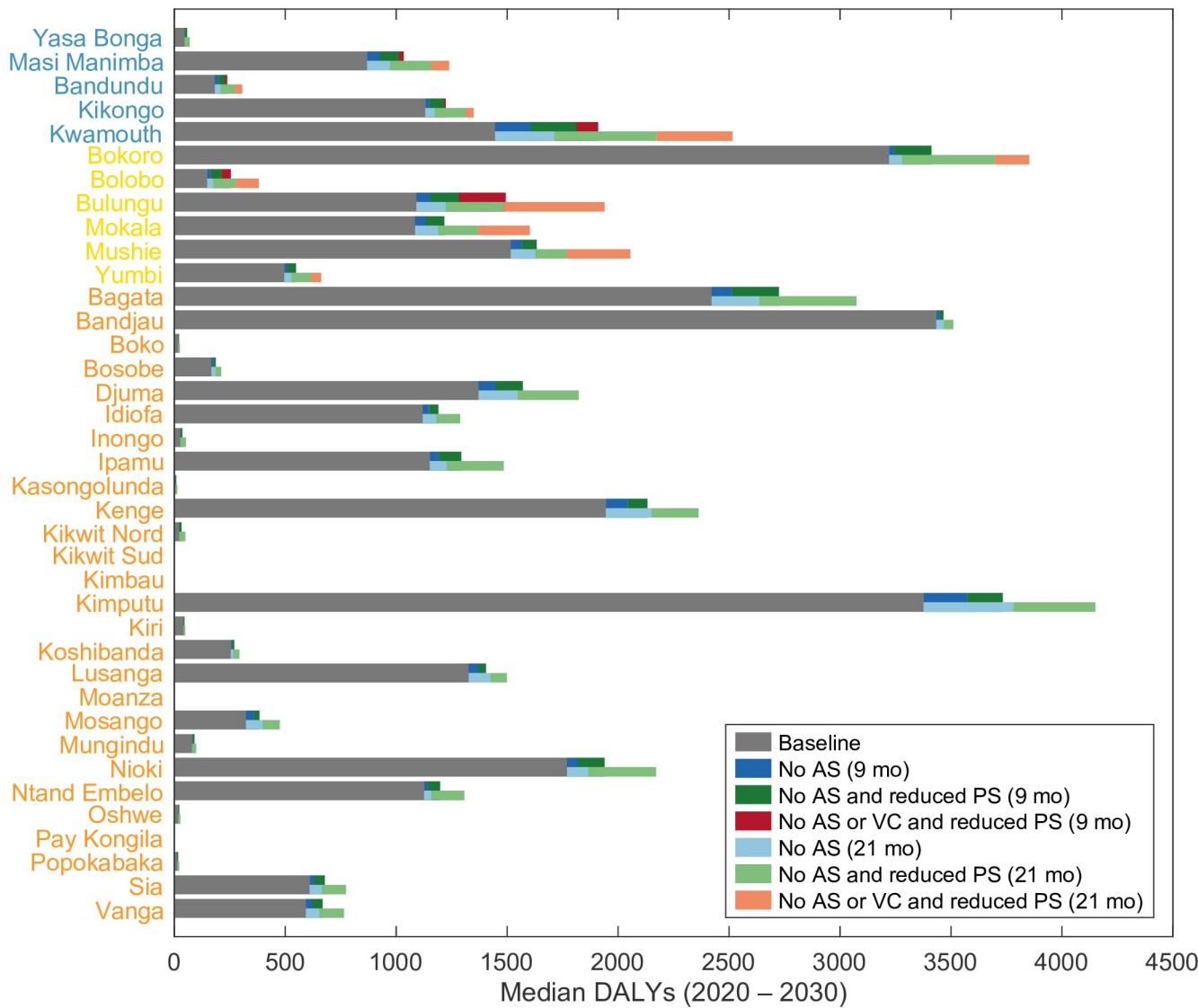

**Fig 6. Stacked bar chart showing DALYs accrued under baseline and interruption scenarios in each health zones of Bandundu Nord and Sud coordinations for the period 2020–2030.** Health zones are colour coded: blue, health zone with existing VC; yellow, health zones with planned VC; orange, health zones without existing or planned VC. The median DALYs (2020–2030) under the baseline strategy. AS: active screening; PS: passive screening; VC: vector control; DALYs: disability-adjusted life years.

the potential pros and cons from a transmission perspective of fexinidazole [39]. They found that reductions in compliance in treatment could lead to lower effectiveness, but there were also the possibility for improved access to treatment and subsequently reduced transmission. Should the oral, single-dose drug, acoziborole, become available following completion of clinical trials [40] and have a suitable safety profile, it may be possible for the gHAT programme to add a "screen and treat" algorithm in the coming years. This approach, which could remove the need for parasitological confirmation prior to treatment, is one way in which the strategies against gHAT could change in the DRC, hopefully enabling more rapid progress towards zero. Other changes to AS or PS algorithms, or operating protocols for VC also have the potential to modify what we describe as our baseline strategy in this study and to accelerate progress.

At present there is limited data for tsetse reduction in four of five health zones with VC prior to 2020 as they have started recently with six-monthly monitoring. As such we assumed 80% tsetse reduction after one year, which is less than the observed 90% in Yasa Bonga health zone and Uganda [28], or 99% in Chad [21, 26] and on par with Boffa in Guinea [27]. Once entomological data from the health zones is available we can appropriately adjust the simulated VC reduction. However previous studies have shown that a reduction in tsetse of > 70% could break transmission [28] and that reductions of this magnitude should be sufficient to interrupt transmission within 4 years [29]. In the DRC Tiny Target deployments are focused on the rivers that serve as the likely tsetse habitats for villages that have reported cases. So, whilst the VC is not exhaustive and is not designed to eliminate tsetse, it is aimed to substantially reduce those populations that are maintaining transmission cycles. Therefore, we expect limited differences in modelled outcomes when replacing the assumed tsetse reduction by observed ones when data becomes available. The levels of protection from VC can be highly variable within a health zone depending on the geographical distribution of Tiny Targets. In this study, we assumed VC covers 100% of population at risk. Other factors such as seasonal distribution of tsetse populations and tsetse reinvasion from connected areas are also out of the scope of this paper. To improve the tsetse dynamics generated by our model, in the future we hope to combine case coverage of VC and observed VC efficiency (within intervention areas) to estimate the overall tsetse reduction in a health zone. Note that simulation results would be very different if a much lower overall reduction percentage (e.g. 20%) were fed into our model, although we believe these low reduction values are very unlikely.

For this analysis we used AS coverage for 2000–2018 but not 2019 onward which could have impacted our results. We had assumed this missing coverage data for 2019 was the mean of 2014–2018. When the data becomes available for the years after 2020 (e.g. until the end of 2022), it would be possible to perform modelling validation to examine our predictions by using the actual screening numbers with the same model fits and compare to the case reporting observed during and post interruption. Our results presented in this article and our open source code mean this validation exercise should be straightforward to conduct, in line with the "testable model outcomes" recommended by the Policy-Relevant Items for Reporting Models in Epidemiology of Neglected Tropical Diseases (PRIME-NTD) checklist [41] (see S3 Text). The medium- to long-term projections could be influenced by model assumptions, however, it would also be possible to refit the model to the actual screening and case data to quantify if there is any statistical evidence of the interruption's impact on PS during 2020. Additionally, we can also factor in some plausible changes such as the decrease in AS participation caused by diminishing awareness of the disease, targeted AS that screens more high-risk individuals, and/or the introduction of new single-dose drug, acoziborole.

## Conclusion

In this study we simulated six scenarios that interrupted gHAT elimination programme activities in the DRC to predict the impact on case reporting, gHAT burden, and time to achieve EoT for the health zones in Bandundu Nord and Sud coordinations. We predicted that EoT could be delayed by a maximum of the length of interruptions in the worst case scenarios (i.e. no AS and reduced detection rate in PS for the interruption period, and, if a health zone had previous or planned VC, an interruption to VC deployments). However, our model estimates the worst case scenarios have an overall 28% increment in the DALYs accrued during 2020–2030 in Bandundu Nord and Sud coordinations, varying from zero in some health zones to 774.2 (Kimputu in the 21-month *No AS and reduced PS* scenario) and 1069.3 (Kwamouth in the 21-month *No AS or VC and reduced PS* scenario) compared to the baseline strategy. The impact on disease burden

depends on endemicity, strength and coverage of medical interventions, and the VC history of the health zone. These results, for a range of health zones of different historical AS coverages and case incidences, align with the previous results presented by Aliee *et al.* in which (using gHAT models developed by two different research groups) there was limited impact from short-term interruptions to gHAT interventions in illustrative "moderate-risk" health zones [20].

The present analysis was the first to examine a potential interruption of VC. We expect VC to have a direct impact on EoT through the reduction in tsetse population which consequently decreases transmission from tsetse to humans. We found a delay in YEoT only occurs when VC is interrupted but it is in general the interruption of PS, not VC, that leads to the most additional DALYs being accrued (see the lengths of the coloured bars for each health zone in Fig 6). In summary, this modelling study indicates that the gHAT elimination programme is fairly robust to short shocks if activities can be resumed again. We believe this is, for the most part, due to the slow-progressing nature of the gHAT infection. Nevertheless, additional disease burden is predicted to arise as a consequence of interruptions and the more interventions are interrupted and the longer they are interrupted for, the more DALYs accumulate.

This and other similar modelling analyses by the HAT MEPP project [17, 29, 33, 42] have been used to support PNLTHA-DRC's quantitative evaluation of their elimination strategy. Modelling cannot perfectly predict the future, and it is acknowledged that financial and operational challenges may sometimes prevent optimal strategies from being conducted. However, now there are a good range of tools to combat gHAT and modelling has allowed the team to see which combinations of interventions are likely sufficient to reach this ambitious goal whilst also considering the efficiency of different approaches in different regions.

## Supporting information

**S1 Text. Methods.** An expansion of the mathematical modelling methodology.
(PDF)

**S2 Text. Additional model outputs.** Individual time series plots for all modelled health zones and additional figures and tables summarising impacts on EoT and DALYs.
(PDF)

**S3 Text. PRIME-NTD criteria.** Addressing the PRIME-NTD criteria for good modelling practises.
(PDF)

## Acknowledgments

The authors thank PNLTHA-DRC for original data collection, and the WHO for data access (in the framework of the WHO HAT Atlas [32]), and Cyrus Sinai and Nicole Hoff from UCLA Fielding School of Public Health for providing health-zone-level shapefiles (current versions can be found at https://data.humdata.org/dataset/drc-health-data). We thank Prof Steve Torr for his feedback on this article.

## Author Contributions

**Conceptualization:** Kat S. Rock.

**Data curation:** Ching-I Huang, Ronald E. Crump, Chansy Shampa, Erick Mwamba Miaka.

**Formal analysis:** Ching-I Huang, Ronald E. Crump, Kat S. Rock.

**Funding acquisition:** Andrew Hope, Erick Mwamba Miaka, Kat S. Rock.

**Investigation:** Ching-I Huang, Ronald E. Crump, Emily H. Crowley, Andrew Hope, Paul R. Bessell, Chansy Shampa, Erick Mwamba Miaka, Kat S. Rock.

**Methodology:** Ching-I Huang.

**Project administration:** Emily H. Crowley, Kat S. Rock.

**Software:** Ching-I Huang, Ronald E. Crump.

**Supervision:** Emily H. Crowley, Kat S. Rock.

**Validation:** Ching-I Huang, Ronald E. Crump, Kat S. Rock.

**Visualization:** Ching-I Huang.

**Writing – original draft:** Ching-I Huang, Emily H. Crowley, Kat S. Rock.

**Writing – review & editing:** Ching-I Huang, Ronald E. Crump, Emily H. Crowley, Andrew Hope, Paul R. Bessell, Chansy Shampa, Erick Mwamba Miaka, Kat S. Rock.

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
