## [Decision Letter · Decision Letter 0]

10 Dec 2022

Dear Huang,

Thank you very much for submitting your manuscript "A modelling assessment of short- and medium-term risks of programme interruptions for gambiense human African trypanosomiasis in the DRC" for consideration at PLOS Neglected Tropical Diseases. As with all papers reviewed by the journal, your manuscript was reviewed by members of the editorial board and by several independent reviewers. The reviewers appreciated the attention to an important topic. Based on the reviews, we are likely to accept this manuscript for publication, providing that you modify the manuscript according to the review recommendations. 

Sincerely,

Enock Matovu

Academic Editor

Victoria Brookes

Section Editor

Reviewer's Responses to Questions

**Key Review Criteria Required for Acceptance?**

**Methods**

-Are the objectives of the study clearly articulated with a clear testable hypothesis stated?

-Is the study design appropriate to address the stated objectives?

-Is the population clearly described and appropriate for the hypothesis being tested?

-Is the sample size sufficient to ensure adequate power to address the hypothesis being tested?

-Were correct statistical analysis used to support conclusions?

-Are there concerns about ethical or regulatory requirements being met?

Reviewer #1: The objectives of the study are not openly stated. Their resemblance simply run as statements in the text. These need to be stated clearly in objectives section to offer the reader with what to expect in the paper.

No hypotheses stated

methods and selected sample size were sufficient and subjected to right statistical analysis.

Model makes use of secondary data majorly. Hence no need for ethical clearance in this case

Reviewer #2: The article is well written and discusses HAT control in the Democratic Republic of Congo during the COVID-19 lockdown.The objectives of this study are clear and the methods to achieve them are well developed. The authors demonstrate a strong mastery of statistical analyzes on modeling. It would be desirable for the authors to refer to the current epidemiological situation of HAT in the Democratic Republic of Congo because the prevalence of the disease has fallen sharply and it is no longer felt in the population as a health problem. This negatively impacts the rate of participation in active screening activities. It would perhaps be desirable for the authors to take this community aspect into account in their model so that it stands out in the analyzes or as limits.

Reviewer #3: Objectives are clearly stated.

Study design is appropriate and has been used previously to address similar objectives.

Study population has been described appropriately.

Appropriate statistical analysis was employed.

No ethical concerns.

**Results**

-Does the analysis presented match the analysis plan?

-Are the results clearly and completely presented?

-Are the figures (Tables, Images) of sufficient quality for clarity?

Reviewer #1: The analysis presented matches the analysis plan. The results are clearly presented. The figures applied are of sufficient quality and quite clear.

Reviewer #2: The results are well presented and clear and seem to me to be in line with the objectives and the method used.

Reviewer #3: Results match the analysis plan as mentioned in the methods section.

Results are clearly presented.

Figures and Images are of sufficient quality for online publication, may need more polishing for printing purposes.

**Conclusions**

-Are the conclusions supported by the data presented?

-Are the limitations of analysis clearly described?

-Do the authors discuss how these data can be helpful to advance our understanding of the topic under study?

-Is public health relevance addressed?

Reviewer #1: There is no conclusion section. This needs to be worked on.

Public health relevance addressed, and authors attempt to discuss the usefulness of the model

Reviewer #2: We would have liked the authors to take into account in the discussion the aspect of the re-invasion of vectors, ... it has been demonstrated on several occasions that if vector control is not concerted and applied in other areas of surrounding health and/or provinces and/or endemic countries, the vectors concerned by the control recolonize the environments and can constitute a non-negligible factor in the resurgence of the disease. Moreover, during the lock-down the care activities did not stop and some mobile teams did little to work. The impact on disease control was therefore low. The aspect on the functionality of the health system is major in the control of HAT in areas with low endemicity. and this was not very elaborated in the discussion. Passive screening can also be impacted by the poor functionality of health centers. Some work has revealed that the rate of use in the two HAT endemic areas may be low and this may make it difficult to implement an effective HAT surveillance system based on passive screening. Sometimes the use of mini teams may be necessary. It is possible that despite the best scenarios even if the patient does not consult the health center, he cannot be diagnosed and treated and remains a danger for the members of his community.

Reviewer #3: The conclusions are supported by the data presented.

No limitations have been explicitly mentioned.

The authors discuss the data and provide insights of its applicability in relation to the topic under study.

Public health relevance is appropriately addressed.

**Editorial and Data Presentation Modifications?**

Reviewer #1: presentation modifications include;

1. Clearly show the key paper headings i.e Introduction, objectives, methodology, findings / results, discussion, conclusion & recommendations. + key guiding assumptions

2. Some important maps for inclusion have been suggested and are highly recommended

3. Model details are missing. Hard to follow and cause duplication.

Reviewer #2: minor revision

Reviewer #3: (No Response)

**Summary and General Comments**

Reviewer #1: 1. What are 'health zones'? need to provide interpretation.

2. Why continuously use the term ‘former Bandundu province? Better to use the current identification.

3. Update your information on countries which have been validated for elimination of gHAT and have since been declared so by WHO in this year 2022.

4. Please state the objectives clearly and even conclusion.

5. Unfortunately, this is not a stand-alone piece of work as there is frequent reference to the ‘previous works’ even for aggregated data used for model fitting. Hence reviewer is compelled to refer to previous team works.

6. Not clearly able to see the disruptive roles of covid, political unrest, disasters disease epidemics etc in the model. Their modes of disruption differ.

7. Active Screening coverage is not uniform due to several reasons. How have you accounted for this?

8. The model was fitted on the 2000-2016 HAT Data yet covid data is only for 2020-2021. How was this harmonized?

9. Vector control is not everywhere in your ‘health zones’ – I imagine so. What is the strength of the model outputs in such a scenario? What population is being effectively protected from tsetse bites by the Tiny targets?

10. Tiny targets are laid along major river banks to trap the riverine glossina species. But it is also true that during the rainy season tsetse flies are more wide spread far beyond the river banks. How have you accounted for this? 

11. Vector control effectiveness has been standardized to 80% across board even where it has never been applied. Under such circumstances am not sure if this VC carries with it any role in the model. Also refer to lines 440 and 441. Why bother with VC? Looks like a redundant model parameter.

12. There was need to set the stage with providing a spatial connectivity for the following; areas being treated with tiny targets (deployment sites) , AS screening sites, PS (health centres), population densities (gridded). This should be map(s).

13. What was particularly unique with Bandundu setting? Any lessons picked? Can this model be automatically duplicated elsewhere?

14. The unique stakeholder inputs are not highlighted anywhere (and in what ways where they involved?)

15. The presence of an open source code is good and could be used for model validation as new actual data becomes available.

Reviewer #2: (No Response)

Reviewer #3: This paper presents a comprehensive exploration of the potential impact of interruptions towards the end of transmission and burden of Gambiense human African trypanosomiasis (gHAT) in thirty-eight health zones in the Democratic Republic of the Congo (DRC). The authors employed an existing deterministic model (implemented in MATLAB) to simulate the transmission and reporting dynamics of gHAT under six scenarios of interruptions to intervention efforts for periods of nine and twenty-one months. The obtained predictions suggest that simultaneous interruption of passive and active screening could substantially increase the burden of gHAT as compared to interrupting active screening alone. 

The paper is well-written, and the presented insights are interesting. However, there are some minor points that need clarification.

1. The specifications of the computational infrastructure platform used to conduct the analysis must be reported indicating CPU/GPU, RAM, OS among others. In addition, the authors should present the runtime analysis, indicating how much compute time is required to run simulations under different interruption scenarios on the employed compute infrastructure. This is important not only for reproducibility purposes but also compute cost estimation for this kind of analysis.

2. Page 6, line 195 and Table 1, column 5; The authors assume “a fixed effectiveness of 80% tsetse reduction in one year - the MEanAS+VC baseline strategy”. The authors need to justify the choice of 80% for this parameter.

3. Model parameters for the simulation are shown, well referenced, and justified. However, it would great to provide a sensitivity analysis of parameters to check the effect of different choice of model parameters to the model predictions.

4. In the discussion section, the authors cite previous work that showed similar predictions between deterministic and stochastic versions of the model. However, it is mentioned that for this work, “a stochastic version of the model could better capture the model”. Why not present the stochastic version of the model? The authors could clarify why a deterministic version is presented if a stochastic version could provide better insights.

PLOS authors have the option to publish the peer review history of their article (what does this mean?). If published, this will include your full peer review and any attached files.

Reviewer #1: No

Reviewer #2: No

Reviewer #3: No

Figure Files:

Data Requirements:

Reproducibility:

References

---

## [Editor Report · Decision Letter 1]

12 Apr 2023

Dear Huang,

We are pleased to inform you that your manuscript 'A modelling assessment of short- and medium-term risks of programme interruptions for gambiense human African trypanosomiasis in the DRC' has been provisionally accepted for publication in PLOS Neglected Tropical Diseases.

Best regards,

Enock Matovu

Academic Editor

Victoria Brookes

Section Editor

---

## [Editor Report · Acceptance letter]

24 Apr 2023

Dear Huang,

We are delighted to inform you that your manuscript, "A modelling assessment of short- and medium-term risks of programme interruptions for gambiense human African trypanosomiasis in the DRC," has been formally accepted for publication in PLOS Neglected Tropical Diseases.

Best regards,

Shaden Kamhawi

co-Editor-in-Chief

Paul Brindley

co-Editor-in-Chief
